# Experimental Study and Performance Characterization of Semi-Flexible Pavements

Guosheng Li [1], Huan Xiong [1,*], Qi Ren [2], Xiaoguang Zheng [2] and Libao Wu [2]

1   Chengdu Xingcheng Construction Management Co., Ltd., Chengdu 610041, China; gsl11041@hotmail.com
2   Shanghai Municipal Engineering Design Institute (Group) Co., Ltd., Shanghai 200092, China; qrendavis@gmail.com (Q.R.); zhengxiaoguang@smedi.com (X.Z.); wulibao@smedi.com (L.W.)
*   Correspondence: hxiong9527@hotmail.com

**Abstract:** Semi-flexible pavement (SFP) is made up of a porous skeleton of asphalt mixture and cement grout. This special structure granted SFP superior strength and durability and made it a promising solution for the paving of heavy trafficked area. This paper performed in-depth study on the mechanistic behavior of SFP. Firstly, the volumetric mix design of SFP was introduced, and followed with strength, moisture susceptibility, viscoelastic behavior, fatigue life as well as rutting resistance through a series of laboratory tests. Marshall stability tests and dynamic stability tests suggested that SFP gained fair strength and rutting resistance from the curing of cement grout. Meanwhile, SFP was found not sensitive to freeze–thaw cycles through indirect tensile tests. In dynamic modulus tests, SFP exhibited significant viscoelastic behaviors. In four-point beam fatigue tests, the average fatigue lives of SFP reached 85.4 k loading repetition under 400 με level. In Hamburg wheel tracking tests, the ultimate rutting depth of SFP was smaller than 2.5 mm. The viscoelastic behavior and rutting propagation of SFP was characterized with master curve and power function by fitting the test results. SFP was also compared with traditional asphalt mixtures in MMLS3 accelerated tests and its performance turned out to be prevailing.

**Keywords:** semi-flexible pavement; Marshall stability; dynamic modulus; four-point beam fatigue test; Hamburg wheel tracking; MMLS3





## 1. Introduction

Semi-flexible pavement (SFP) is a composite structure of open-graded asphalt mixture filled with cement grout. It was firstly proposed by French engineers in the 1960s, known as Salviacim [1]. Starting from 1970s, this technique was further utilized in airfields across Europe and North America with extensive studies on its mechanical properties [2]. During this period, researchers referred to this material as resin modified pavement [3] or Densiphalt [4]. With better knowledge of its performance, subsequent researchers in recent decades preferred the acronym "SFP" [5] to highlight its semi-flexible behavior or "grouted macadam" [6] to emphasize its composition. Despite its various names in history, the fundamental idea of filling the air voids in porous asphalt mixture with cement grout remained consistent [7].

SFP was initially designed as an alternative solution to concrete pavement due to its jointless feature as well as lower cost. Early applications in airfield pavements were positively reported [8]. The heavy and channelized load caused by aircraft wheels was challenging to most asphalt pavements, but SFP was able to maintain fair rutting resistance, skid resistance and durability. The superior performance soon received worldwide recognition. The application of SFP was thus expanded to bridge deck pavements [9], heavy trafficked intersections [10], and bus rapid transit systems [11].

The material design of SFP included two key elements, open-graded asphalt mixture and cement grout. The open-graded asphalt mixture was supposed to possess a void ratio of at least 25% [12], in order to permit the cement infiltration. The design of cement

grout was supposed to maintain a balance between fluidity and strength [13]. In recent years, innovative techniques and methodologies have been frequently used in SFP design. Husain et al. [14] studied the effects of aggregate gradation and concluded that air voids in asphalt matrix would positively contribute to the compressive strength of SFP. This behavior was contrary to traditional asphalt pavements. As for moisture susceptibility concerns, Fakhri and Mottahed [15] studied the performance of warm mixed asphalt mixture containing RAP and nanoclay, which employed a similar mechanism with SFP. The conclusion suggested that enhancing the strength of asphalt mixture could also result in better moisture performance. A comparative study of rutting resistance and moisture damage test methods was carried out by Fakhri et al. [16]. Fair correlation was found between deformation strength, dynamic creep test, and wheel tracking results. Wang et al. [17] added carboxyl latex into cement grout and used a warm-mix technique in open-graded asphalt mixture. Modified SFPs were evaluated with various tests and compared with unmodified specimens. The comparison suggested that carboxyl latex could effectively improve rutting resistance as well as fatigue lives. Cai et al. [18] used nanoindentation, scanning electron microscopy as well as energy dispersive spectroscopy to identify the microstructural characteristics of SFP. It was found that the increase in grout material would further enhance the bonding between asphalt and cement. Cai et al. [19] evaluated the interlocking of SFP with X-ray computed tomography. The work proposed a classification method by identifying the effective contacts between aggregates and cement grout, and concluded that only a limited number of contacts were helpful to the bonding.

Despite the extensive research on the material properties of SFP, its structural design has not been thoroughly studied yet. Oliveira et al. [20] investigated the fatigue life of SFP with two comparative approaches. The empirical fatigue threshold was found conservative and thus damage accumulation was necessary to make more accurate estimations. Bharath et al. [21] used cored specimens from SFP test sections to evaluate the mechanical properties. This study employed indirect tensile strength and resilient modulus tests, and highly commended the performance of SFP. Skid resistance and rutting depth were also recorded in subsequent observations and were found satisfactory. Zhang et al. [22] conducted performance tests with indirect tensile approach. The dynamic modulus of SFP was found higher than that of traditional asphalt mixture. Fair resistance to rutting and moisture damage was also observed. Despite the efforts, it should also be noted that a standard test-characterization-prediction procedure for SFP was still absent. Although Mechanistic-Empirical Pavement Design Guide (MEPDG) was introduced two decades ago [23], SFP has not been incorporated into the framework yet.

Based on previous research, there were still a few concerns regarding the SFP performance and modeling. This paper intended to perform an in-depth investigation in terms of the mechanistic behavior as well as model characterization of SFP. A series of laboratory tests were conducted to address multiple concerns as listed in Table 1. Tests 1 to 4 were strength/stability-based tests, which were used to validate the mix design and provide a fundamental understanding of the SFP material. Tests 5 to 8 were more advanced performance-oriented tests, aiming to quantitatively characterize the mechanistic behavior under simulated traffic loading. The study of this work was used to support the pavement construction of the latest Chengdu Bus Rapid Transit stations.

**Table 1.** Performance test of SFP and the major concerns.

| No. | Test Names | Concerns |
|---|---|---|
| 1 | Marshall Stability | Influence of cement curing |
| 2 | Dynamic Stability | Rutting resistance |
| 3 | Indirect tensile strength | Freeze–thaw cycles |
| 4 | Flexural Strength | Flexural strength |
| 5 | Dynamic modulus | Viscoelastic behavior |
| 6 | Four-point beam fatigue | Fatigue life |
| 7 | Hamburg wheel tracking | Rutting propagation |
| 8 | MMLS3 accelerated wheel tracking | Comparison with asphalt mixtures |

## 2. Materials Design

The mix design of SFP specimen consisted of 3 steps. Firstly, an open-graded asphalt mixture was designed using volumetric method with a gyratory compactor. Secondly, the water–cement ratio of cement grout was determined based on strength, fluidity, shrinkage rate as well as bleeding rate. Lastly, the strength and stability of SFP must be tested through a series of tests to validate the mix design.

### 2.1. Open-Graded Asphalt Mixture

Gradation was a primary concern in the mix design of asphalt mixture. In this study, 3 different gradations were prepared, as shown in Figure 1. The first and second gradation were close to the asphalt mixture commonly used in permeable pavements. The tested void ratios of the fabricated mix were 23% and 19%, respectively. Since the void ratios were not satisfactory for SFP mix, the final gradation was further adjusted to achieve a void ratio of 25%.

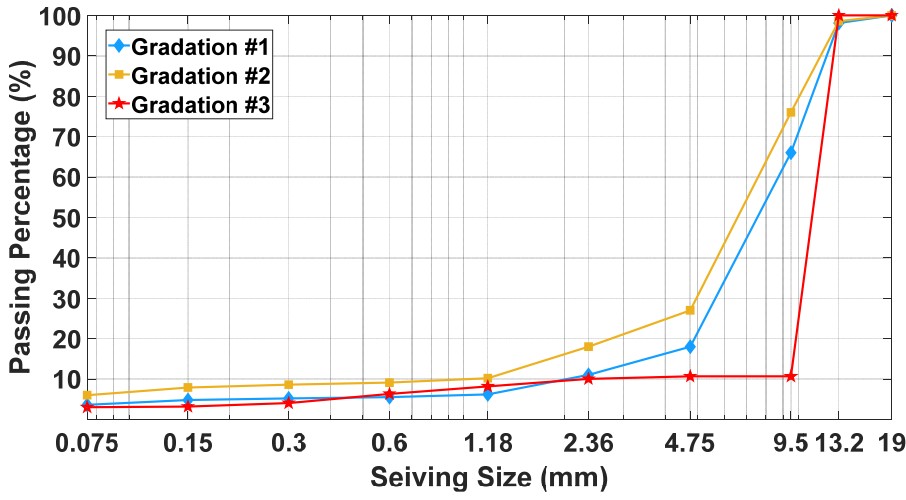

**Figure 1.** Gradation of open-graded asphalt mixture. Void ratios are 23%, 19% and 25% for #1, #2 and #3, respectively.

A high-viscosity modified asphalt was used to prepare the open-graded asphalt mixture using volumetric design method. The penetration was 68 (0.1 mm) and viscosity was 70,000 Pa·S based on vacuum capillary viscometer test. The asphalt content was determined by a group of Marshall stability tests and eventually adjusted to 2.5%. Cylinder specimens were prepared for dynamic modulus tests and Hamburg wheel tracking. As for four-point beam fatigue tests, slab specimens with the same gradation and asphalt content were compacted with a roller.

### 2.2. Cement Grout

For cement-based material, the water–cement ratio was a key parameter in the design. The rise of this ratio would result in a lower strength but better workability, i.e., easier to flow and more convenience in the construction phase. Thus, an ideal water–cement ratio should be able to balance the concerns of strength as well as workability. It was selected based on flexural strength, compressive strength, fluidity, shrinkage rate and bleeding rate. The strength test results were shown in Figure 2.

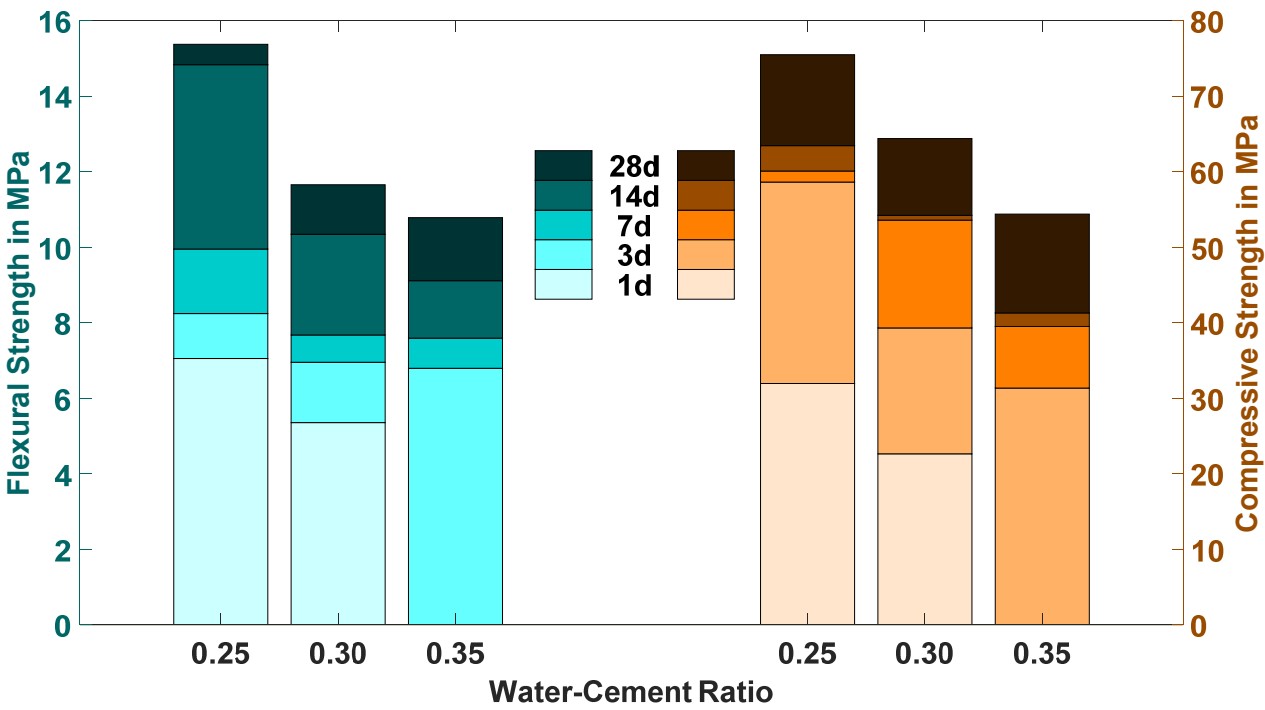

**Figure 2.** Flexural strength and compressive strength of cement grout given different curing times.

For the cases of 0.35 water–cement ratio, the specimen cannot cure in 1 d. It could be seen that the impact of water–cement ratio was obvious. Both flexural strength and compressive strength increased significantly with curing time. The difference increasing rate also suggested certain anisotropy. As water–cement ratio raised from 0.25 to 0.35, the 28 d-strength would be decreased by 30% and 28% percent in flexural mode and compressive mode, respectively. The tradeoff between strength and workability needed to be further investigated through a series of tests on fluidity, shrinkage rate and bleeding rate, as shown in Table 2.

**Table 2.** Fluidity, shrinkage rate and bleeding rate at different w/c ratios.

| W/C | Fluidity in s | Shrinkage Rate in % | Bleeding Rate in % |
|---|---|---|---|
| 0.25 | 11.2 | 0.10 | 1.8 |
| 0.30 | 9.3 | 0.15 | 2.1 |
| 0.35 | 8.5 | 0.28 | 2.5 |

The final water–cement ratio was set to 0.25 to balance strength and workability of cement grout. After the open-graded asphalt mixture was fabricated and cooled, the specimens were covered in a plastic membrane and placed in a cylinder or rectangular mold. The cement grout would be poured from the top surface to form SFP specimens, as shown in Figure 3.

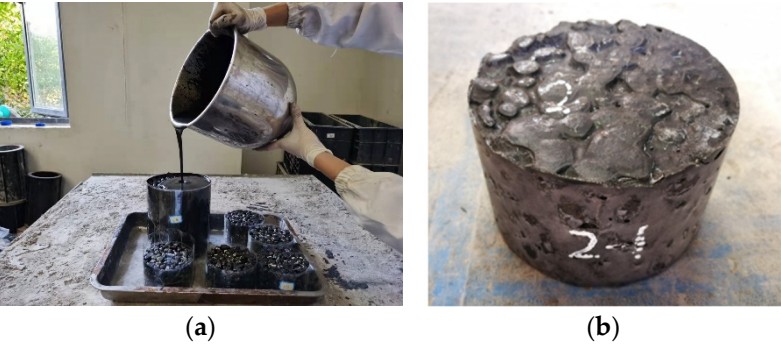

|  (**a**)  |  (**b**)  |

**Figure 3.** SFP specimen fabrication: (**a**) filling cement grout; (**b**) cured SFP specimen.

### 2.3. Mix Design Validation

The designed SFP specimens were validated through a series of tests to investigate the compressive strength, flexural strength and stability, including Marshall stability tests, dynamic stability tests, freeze–thaw cycles and fracture strength tests.

Marshall stability test was a classical method to evaluate the stability of HMA specimens. A compressive load would be applied from the lateral direction of the cylinder at a constant loading rate of 50.8 mm/min till failure. For SFP specimens, this simple method could be used to quantify the effect of cement curing. Five groups of specimens were tested. The first group was tested before cement grout was filled. The second and third group was cured for 3 and 7 days, respectively. The fourth group was tested after 7 days curing and 48 h immersion to examine moisture susceptibility. The last group was tested after 28 days curing, but the strength exceeded the range of the equipment. The test results were plotted in Figure 4.

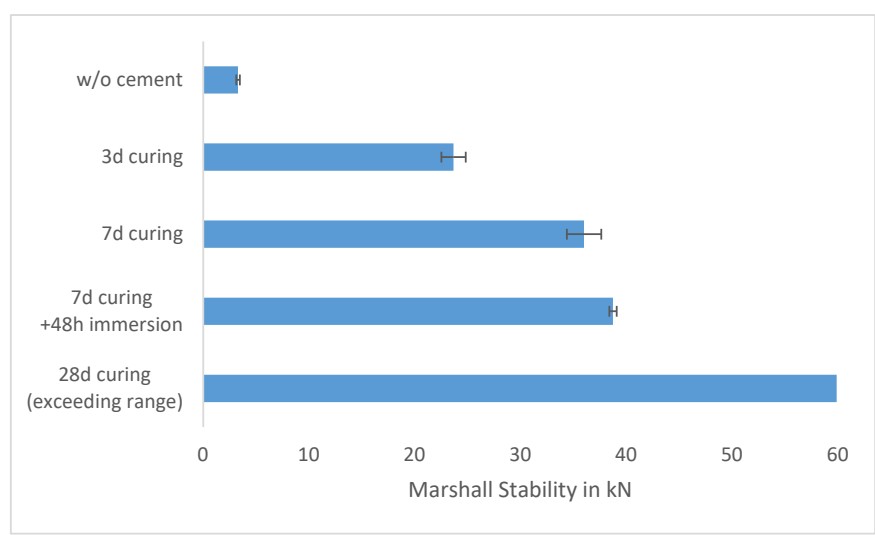

**Figure 4.** Marshall stability test results at different curing times.

The Marshall stability increased along with the curing time. After 3 d curing, it increased approximately 7 times compared with the original open-graded asphalt mixture. In the fourth group, the stability further increased after immersion in water, which suggested that the cement was further cured. The moisture susceptibility was proofed satisfactorily. As for the last group, the 60 kN result represented the maximum value the equipment could measure, so the exact stability remained unknown. In contrast, the Marshall stability of 2 commonly used HMA in China, SMA-13 and AC-20, were both smaller than 20 kN. It should also be noted that Marshall stability test was not a feasible test method for rigorous

study of the mechanistic behavior of SFP. In this study, it served as a tentative attempt to capture the influence of cement curing.

Dynamic stability was an efficient approach to study the rutting resistance from qualitative level. A slab SFP specimen would be loaded with a rubber wheel. The tire pressure was set to 0.7 MPa to simulate traffic load. The loading times needed to cause 1 mm rut depth was recorded as dynamic stability. For SFP specimens, this method was used to study the effect of curing as well as loading temperature. Four groups of dynamic stability tests were conducted. The test results were shown in Figure 5.

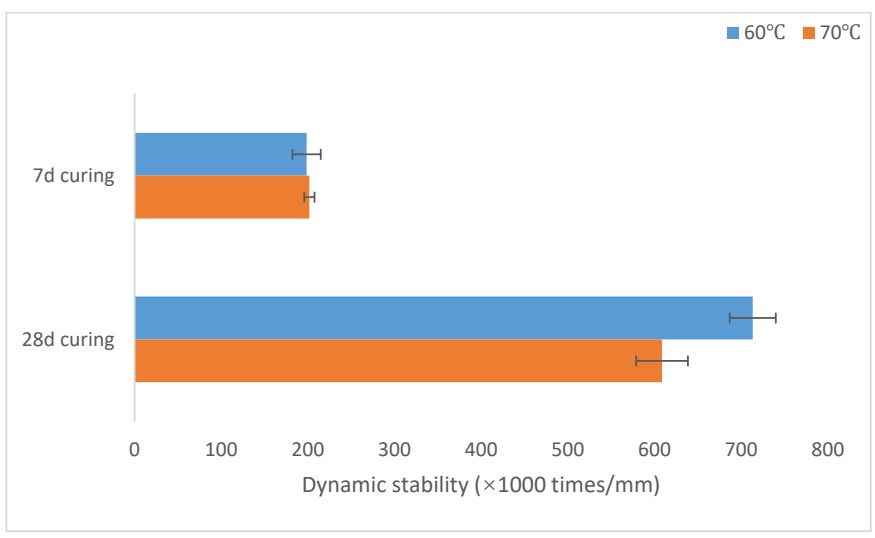

**Figure 5.** Dynamic stability test results at different test temperature and curing times.

It could be seen that the curing time could greatly affect the dynamic stability of SFP. The rutting resistance of 28 d-curing specimens were 259% and 202% percent higher than that of 7 d-curing specimens. The improvement of rutting resistance was supposed to be caused by the curing of cement grout. The dynamic stability of SFP material was about 10 times greater than commonly used asphalt mixtures.

As for indirect tension test, a vertical load would be applied from the top and cause tension in the center of the specimen. The test was in strain-controlled mode and the maximum tensile stress was recorded as the strength. The indirect tensile strength and void ratio variation were tested after freeze–thaw cycles. The test results were shown in Figure 6.

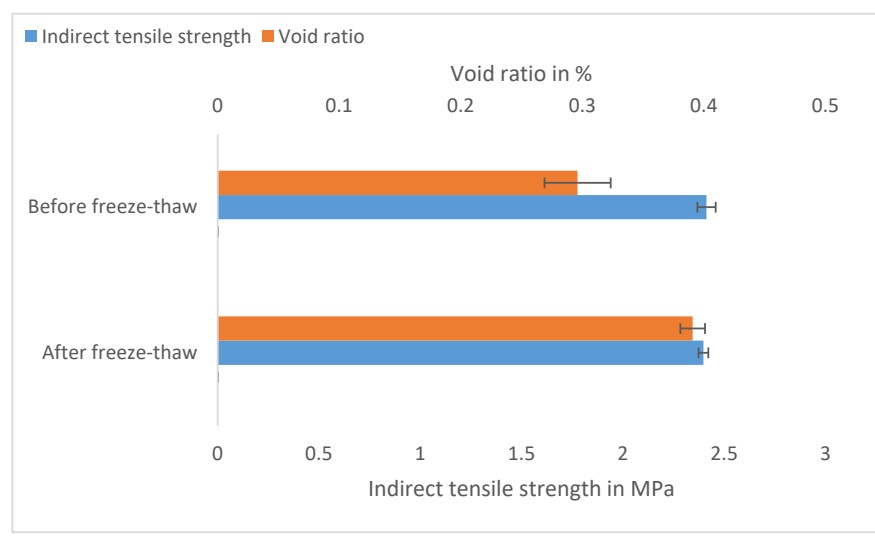

**Figure 6.** Indirect tensile strength and void ratio changes after freeze–thaw cycles.

The indirect tensile strength of SFP specimens after freeze–thaw cycles did not significantly decrease. The specimens maintained approximately 99.4% tensile strength. Meanwhile, the void ratio increased from 0.296% to 0.391%. The test results suggested that SFP material could be regarded as waterproof.

The flexural strength test was performed at −10 °C to study the influence of low temperature conditions. The size of the beam specimen was 250 mm × 35 mm × 30 mm. The specimen was supported at a span of 200 mm and loaded from the center at a constant rate of 50 mm/min till failure. In total 7 SFP specimens were tested. The flexural strength was tabulated along with their failure strains in Table 3.

**Table 3.** Statistical results of flexural strength tests.

| - | Flexural Strength in MPa | Failure Strain in $\mu\varepsilon$ |
|---|---|---|
| Average | 11.18 | 2233 |
| Median | 11.16 | 2283 |
| 75% quantile | 9.95 | 1983 |
| Standard deviation | 0.87 | 217 |

The average flexural strength and failure strain of SFP specimens reached 11.18 MPa and 2233 $\mu\varepsilon$, respectively. The flexural strength range of dense graded asphalt was between 4.5 and 7.5 MPa, according to Cui et al. [24], depending on the angularity and sphericity. Obviously, SFP possessed significantly greater strength.

## 3. Performance Tests

### 3.1. Performance Test Method

As a composite material of asphalt mixture and cement, the application of SFP in pavements involved the following concerns:

(1)   Did viscoelasticity exist in SFP since open-graded asphalt mixture was part of its composition?
(2)   Under repeated loading, what was the expected fatigue life and rutting depth of SFP?
(3)   How good was the performance of SFP compared with other asphalt mixture?

The strength and stability tests conducted in Section 2 was not sufficient to address these issues, so a few more advanced test methods were needed. In this section, dynamic modulus test was used to study the viscoelastic behavior by applying multi-frequency loads at different temperatures, and a sigmoidal master curve would be fitted to characterize the modulus variation under these conditions. Fatigue life and rutting depth would be studied and characterized by four-point beam fatigue test and Hamburg wheel tracking test, respectively. In the end, a direct comparison between SFP and other commonly used asphalt materials was made through a one-third scale accelerated loading test using Model Mobile Load Simulator (MMLS3). The tests setup was shown in Figure 7.

A dynamic modulus test was conducted using a universal testing machine (UTM) in accordance with AASHTO T 342 [25], as shown in Figure 7a. A cylindrical SFP specimen with a diameter of 100 mm and height of 150 mm was subjected to a sinuous load with a series of frequencies of 0.1, 0.5, 1, 5, 10 and 25 Hz. The deformation of the central 70 mm was measured with three linear variable differential transformers (LVDTs). The test temperature was set to 20, 40 and 60 °C.

Four-point beam fatigue tests were employed to study the fatigue resistance of SFP material. The specimen preparation and test equipment were shown in Figure 7b. The size of the specimens was 380 mm × 50 mm × 63.5 mm. The curing time was 28 days. The test temperature was set to 15 °C and loading frequency was 10 Hz. The specimen was loaded at a constant strain of 400 $\mu\varepsilon$.

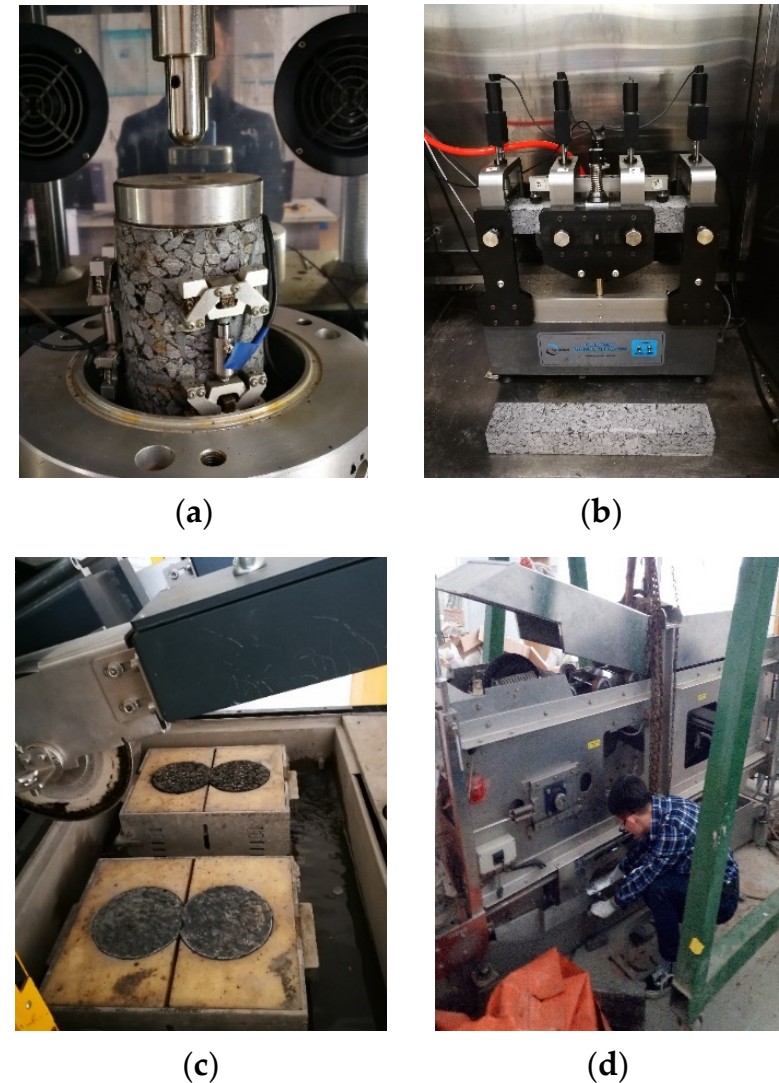

**Figure 7.** Performance tests of SFP. (**a**) Dynamic modulus test; (**b**) four-point beam fatigue test; (**c**) Hamburg wheel tracking test; (**d**) MMLS3 accelerated wheel tracking test.

The rutting resistance of SFP was evaluated using Hamburg wheel tracking (HWT) test. For HWT tests, 150 mm × 60 mm cylinder specimens were fabricated with gyratory compactor. The test condition was 50 °C and water bath. The load was applied with a steel wheel with a magnitude of 0.7 MPa pressure. The test would terminate when rutting depth was greater than 10 mm or the loading repetition reached 20,000 times The HWT apparatus and specimens were shown in Figure 7c.

MMLS3 accelerated test was used to compare the rutting propagation between SFP specimens and typical asphalt mixtures. The advantage of this device was validated by Epps et al. [26] and Lee et al. [27]. The MMLS3 device and test setup were shown in Figure 7d. The size of the specimens was 900 mm × 200 mm with a thickness of 100 mm. The specimen would be bathed in 20 °C air during the test period. The loading system would apply channelized loads with rubber tires at a magnitude of 50 kN and tire pressure of 700 kPa to simulate real traffic loads. The loading frequency was 5000 times/h to accelerate rutting propagation.

*3.2. Test Results and Analysis*

3.2.1. Dynamic Modulus Test

The test results of dynamic modulus tests were plotted in Figure 8.

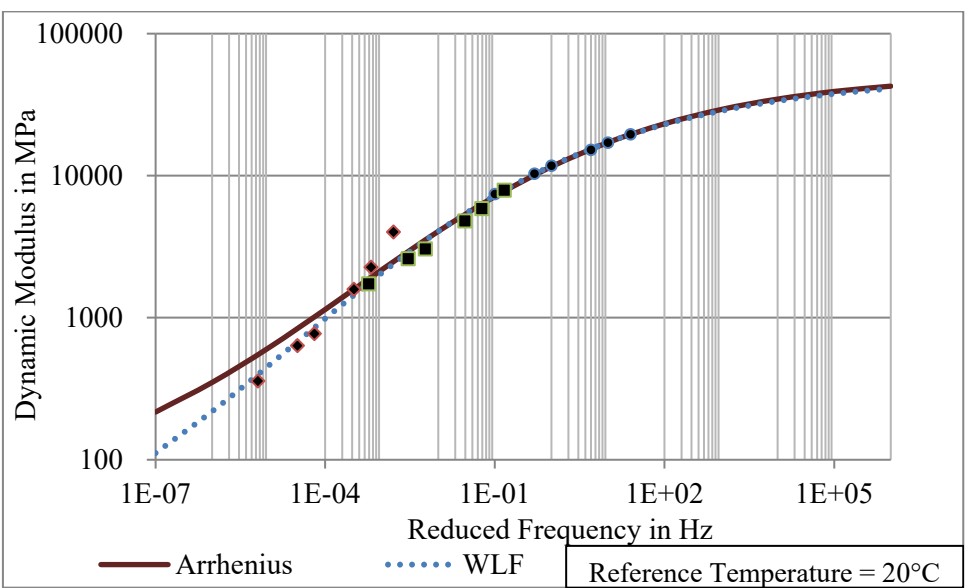

**Figure 8.** Dynamic modulus test results and fitted master curve with WLF and Arrhenius equation.

To characterize the viscoelastic behavior of SFP, a sigmoidal master curve was fitted using two approaches, known as the WLF method and the Arrhenius equation [28]. The master curve could be expressed as Equation (1).

$$\log E* = \delta + \frac{\alpha}{1 + \exp(\beta + \gamma \log(f_r))} \tag{1}$$

where, $\delta$, $\alpha$, $\beta$, $\gamma$ were parameters from regression,
  $E*$ was the dynamic modulus in MPa,
  $f_r$ was the reduced frequency, which could be calculated as Equation (2).

$$\log(f_r) = \log(f) + a_T \tag{2}$$

where, $f$ was the loading frequency in dynamic modulus test,
  $a_T$ was the shift factor depending on the test temperatures, which could be calculated as Equations (3) and (4). For WLF method, the shift factor could be calculated as

$$\log(a_T) = \frac{-C_1(T - T_r)}{C_2 + T - T_r} \tag{3}$$

where, $T$ was the test temperature in Celsius,
  $T_r$ was the reference temperature in Celsius, which was set to 20 °C,
  $C_1$, $C_2$ were regression parameters.
  Alternatively, the shift factor could be calculated using Arrhenius Equation as

$$\log(a_T) = \frac{E_a}{\ln(10) \times R}\left(\frac{1}{T} - \frac{1}{T_r}\right) \tag{4}$$

where, $E_a$ and $R$ could be acquired from regression,
  $T$ and $T_r$ were the test temperature and reference temperature in Kelvin.
  It could be inferred that SFP specimens exhibited obvious viscoelastic behavior. The dynamic modulus of SFP materials could be properly predicted by the master curve fitted either with the WLF method or Arrhenius equation. The master curves were close to each other in high frequency but differs in low frequency. The fitted parameters were tabulated in Table 4.

**Table 4.** Fitted parameters and adjust $R^2$ of master curves.

| Shift Method | WLF | | Arrhenius Equation | |
|---|---|---|---|---|
| | $\delta$ | 0.9465 | $\delta$ | 1.6874 |
| | $\alpha$ | 3.7549 | $\alpha$ | 3.0311 |
| Fitting Parameters | $\beta$ | −1.5970 | $\beta$ | −1.2871 |
| | $\gamma$ | −0.3539 | $\gamma$ | −0.3698 |
| | $C_1$ | 8.1560 | $E_a$ | 68,415.9120 |
| | $C_2$ | 48.6732 | $R$ | 2.9042 |
| Adjusted $R^2$ | 0.9973 | | 0.9939 | |

The fitting results suggested that the adjusted $R^2$ was close to 1, and thus the master curves fitted from both methods were able to explain the modulus variation with time and loading frequency. The viscoelastic behavior of SFP material could also be validated by the modulus-phase angle plot shown in Figure 9, known as black plot.

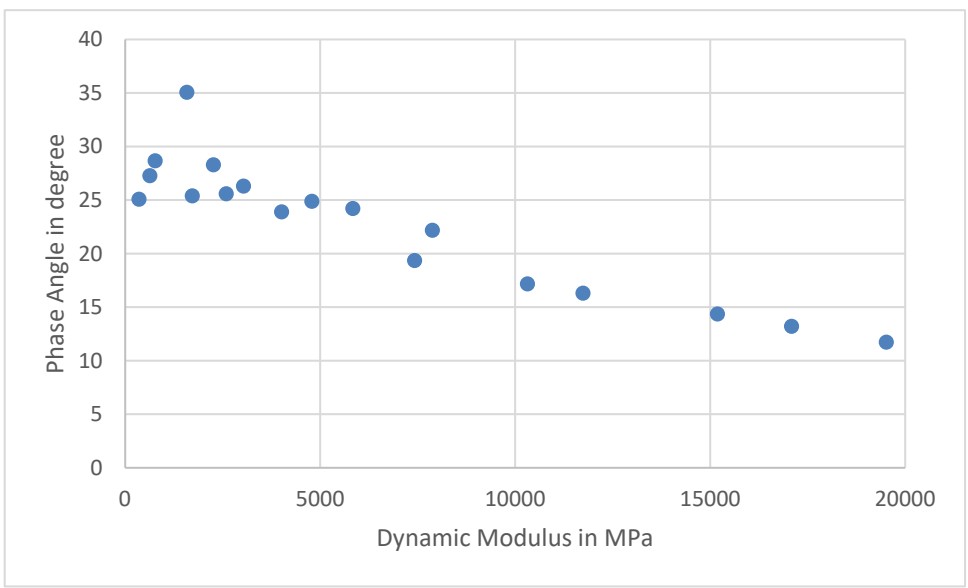

**Figure 9.** Black plot of SFP specimen (dynamic modulus vs. phase angle).

It could be seen that the phase angle would basically decrease with the increase in dynamic modulus. SFP tended to be more elastic at high frequencies and more viscous at low frequencies.

3.2.2. Four-Point Beam Fatigue Test

The test results of eight specimens were summarized in Table 5 along with certain statistical analysis. Despite the significant variance, the test results did not reject normality hypothesis, which suggested that the fatigue lives of SFP specimens could be described by a normal distribution. The test results suggested that the average fatigue lives of SFP reached 85.4 k loading repetition, which was obviously higher than open-graded asphalt mixture reported by literature [29].

However, the comparison between SFP and dense-graded asphalt mixture could be tricky. Due to the introduction of cement, the specimen became less flexible. Under strain-controlled mode, if the strain level was high enough, the fatigue life of SFP would be shorter than dense-graded asphalt mixture, which was quite misleading. Thus, to make this comparison justifiable, stress-controlled mode should be employed, which would be extremely time consuming. Alternatively, such comparison should be made through a wheel tracking test, which was discussed in Section 3.2.4.

**Table 5.** Four-point beam fatigue test results and statistical normality tests.

| - | Average | 25% Percentile | Median | 75% Percentile | Standard Deviation |
|---|---|---|---|---|---|
| Fatigue life (×1000) | 85.4 | 40.7 | 91.9 | 127.4 | 50.5 |
| Normality test | | Anderson-Darling test | | Kolmogorov–Smirnov test | |
| *p* value (5% significance level) | | 0.9026 | | 0.9552 | |
| Reject normality? | | No | | No | |

### 3.2.3. Hamburg Wheel Tracking (HWT) Test

The test results of HWT test results were shown in Figures 10 and 11 along with the fitting results.

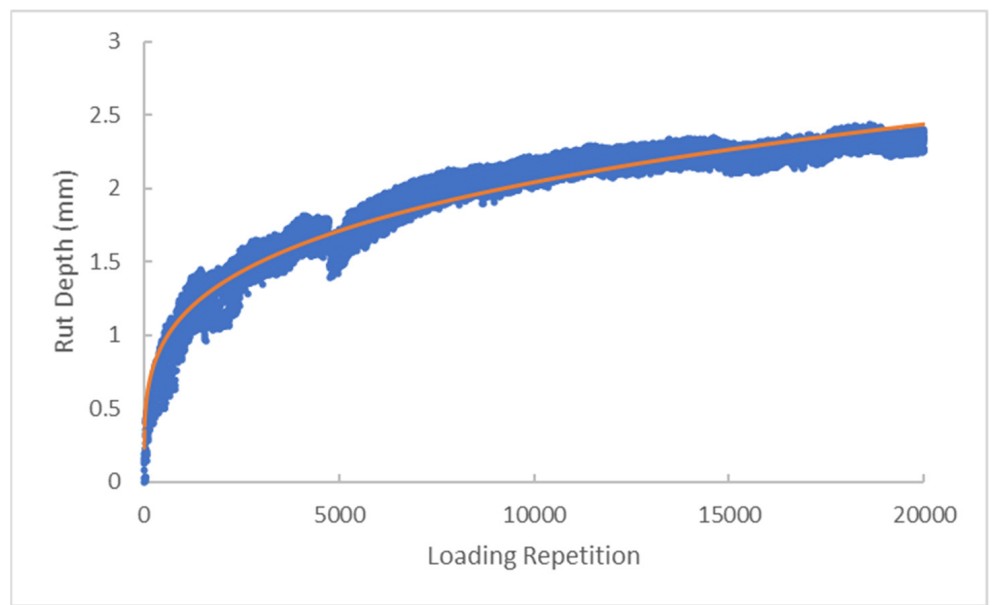

**Figure 10.** 1# HWT test results and fitted power model.

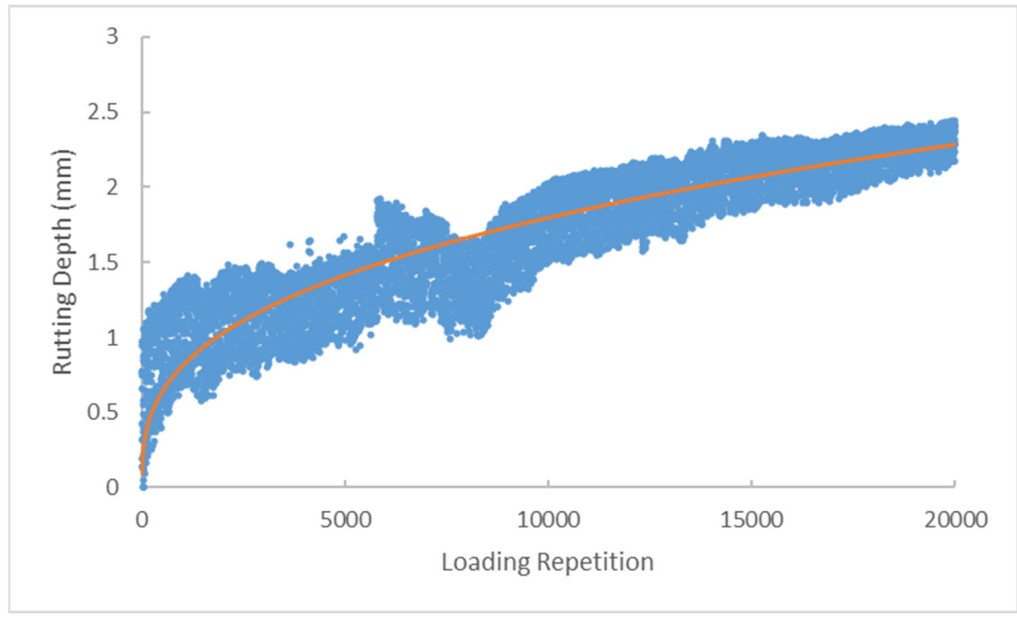

**Figure 11.** 2# HWT test results and fitted power model.

The correlation between rutting depth and loading repetition could be characterized with a power equation as

$$d_p = a \times N^b \tag{5}$$

where, $d_p$ was the cumulative rutting depth in mm,

$N$ was the number of loading repetitions,

$a$, $b$ were regression parameters.

The fitting parameters were tabulated in Table 6.

**Table 6.** The regression results of HWT test.

| Parameters | HWT Test 1# | HWT Test 2# |
|---|---|---|
| a | 0.195 | 0.075 |
| b | 0.255 | 0.345 |
| Adjusted $R^2$ | 0.94 | 0.82 |

It could be seen that the rutting depth was smaller than 2.5 mm at 20,000 loading repetition. Considering the fact that the specimens were water bathed at 50 °C, the moisture susceptibility and rutting resistance of SFP was satisfactory. Meanwhile, the fitted curves maintained $R^2$ of 0.94 and 0.82 for each HWT test results. It could be inferred that the power model could be used to predict the rutting propagation of SFP materials.

### 3.2.4. MMLS3 Accelerated Pavement Test

Four different structures were tested, as summarized in Table 7, along with their mix types and nominal maximum aggregate sizes. The first structure was composed of gap-graded stone mastic asphalt (SMA) and dense graded asphalt concrete (AC), which was a commonly used structure in pavement construction. The second structure was made up of two 5 cm-SMA layers, and high-viscosity modified asphalt was used to enhance its rutting resistance. To investigate the performance of SFP, the SMA used in the top layer of the first structure and bottom layer of the second structure were replaced with SFP.

**Table 7.** Four structures tested in MMLS3 accelerated wheel tracking test.

| Structure No. | Top Layer | Bottom Layer |
|---|---|---|
| 1 | 4 cm SMA-13 | 6 cm AC-20 |
| 2 | 5 cm SMA-13 (High viscosity modified asphalt) | 5 cm SMA-13 (High viscosity modified asphalt) |
| 3 | 4 cm SFP-13 | 6 cm AC-20 |
| 4 | 5 cm SMA-13 (High viscosity modified asphalt) | 5 cm SFP-13 |

The test results of MMLS3 accelerated tests were plotted in Figure 12. No cracking or stripping was observed during the test. The rutting depths were recorded after 1 thousand, 5 thousand, 10 thousand, 100 thousand, 300 thousand, 500 thousand, 700 thousand and 1 million loading repetitions. It could be seen that the rutting propagation was greatly delayed when SFP was introduced in the 3rd and 4th structure. The ultimate rutting depth of the 3rd structure was only 1/3 of that of the 1st structure. Meanwhile, it could be inferred that using SFP as the surface layer would better contribute to the rutting performance by comparing the ultimate rutting depth of these structures.

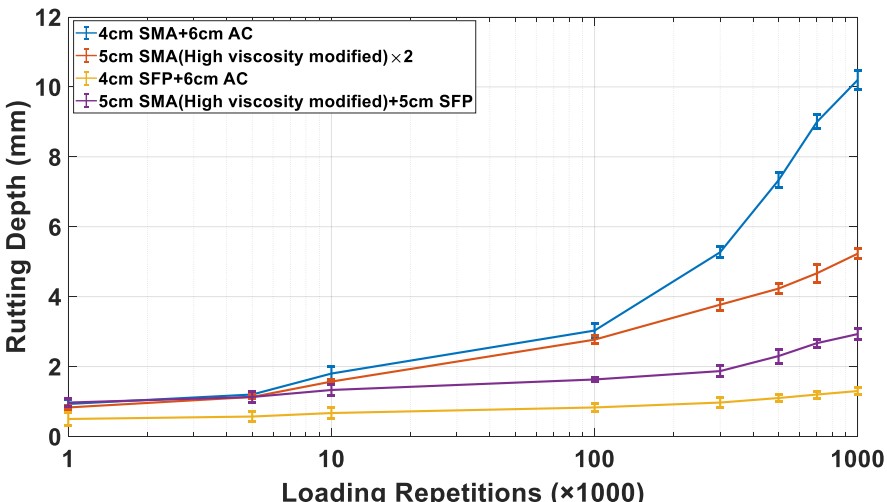

**Figure 12.** MMLS3 accelerated wheel tracking test results, loading repetitions are in log scale.

## 4. Summary

SFP material have been considered as a promising solution for heavily trafficked pavements since the last century. In this paper, a series of performance tests were conducted to study its mechanical strength as well as performance under simulated traffic loads. Compared with previous research, SFP was thoroughly studied in a quantitative manner. Mechanistic models were fitted based on test results to better characterize and predict the performance. The major findings were summarized as follows.

1.  The SFP material consisted of two parts, open-graded asphalt mixture and cement grout. The mix design of open-graded asphalt mixture could be accomplished with volumetric method, and cement grout should be designed based on strength, fluidity, shrinkage rate and bleeding rate. Thus, two critical parameters in material design were target void ratio and water–cement ratio, which were 25% and 0.25 in this study, respectively. These two parameters needed to be chosen carefully to ensure the performance and durability of SFP material.

2.  A series of strength and stability tests were conducted to validate the mix design of SFP. The Marshall stability test suggested that the strength of SFP specimens increased approximately 7 times compared with the original open-graded asphalt mixture. The Marshall stability could raise to 35 kN after 7 days curing, which was significantly higher than HMA. The dynamic stability test results suggested that the rutting resistance of SFP would be greatly affected by the curing time. The rutting resistance of 28 d-curing specimens would be 202% to 259% higher than that of 7 d-curing specimens. This implied that the rutting resistance was actually provided by the cement grout instead of asphalt mixture. The indirect tensile strength suggested that SFP specimens were able to maintain 99.4% tensile strength after freeze–thaw cycles. Meanwhile, the void ratio barely changed so it could be regarded as waterproof. The flexural Strength of SFP could reach 10 MPa, which was believed to be satisfactory for heavy traffic.

3.  Viscoelastic behavior of SFP was observed in dynamic modulus tests. Two master curves were fitted using WLF and Arrhenius equation, respectively. The master curves were mostly identical except for the low frequencies smaller than $10^{-4}$ Hz. The adjusted $R^2$ suggested both of the master curves could characterize the modulus variation of SFP material under different temperatures and loading frequencies.

4.  Strain-controlled four-point beam fatigue test was employed to study the fatigue resistance of SFP. The test results suggested that the average fatigue life of SFP under 400 $\mu\varepsilon$ reached 85.4 k loading repetitions. As for the characterization of rutting resistance, a Hamburg wheel tracking test was used and a power model was fitted

based on tested data. The ultimate rutting depth after 20,000 times loading repetition maintained below 2.5 mm and tended to be convergent.

5.  Four different structures were tested with a MMLS3 accelerated testing device to compare the performance between SFP and other commonly used asphalt mixtures. It was found that the rutting propagation could be greatly stemmed if SFP was introduced into a typical HMA structure. Meanwhile, placing SFP on the top layer of certain structure turned out to be more effective in design.

To conclude, the overall performance of SFP material was believed to be superior, compared with traditional asphalt mixtures. Future research should focus on the engineering practice of SFP and its long-term performance in the field.

**Author Contributions:** Conceptualization, G.L. and H.X.; methodology, G.L.; validation, H.X., Q.R. and X.Z.; formal analysis, Q.R.; investigation, L.W.; resources, G.L.; data curation, X.Z.; writing—original draft preparation, G.L.; writing—review and editing, H.X.; visualization, Q.R.; supervision, X.Z.; project administration, G.L.; funding acquisition, G.L. All authors have read and agreed to the published version of the manuscript.

**Funding:** This research was funded by Chengdu Xingcheng Construction Management Co. Ltd., grant number K2020K180B-01. The APC was funded by Shanghai Municipal Engineering Design Institute (Group) Co. Ltd.

**Institutional Review Board Statement:** Not applicable.

**Informed Consent Statement:** Not applicable.

**Data Availability Statement:** The data presented in this study are available on request from the corresponding author.

**Acknowledgments:** The authors would acknowledge the Key laboratory of Road and Traffic Engineering of the Ministry of Education at Tongji University for the support in laboratory tests.

**Conflicts of Interest:** The authors declare no conflict of interest.

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
