# Peer review of "Experimental Study and Performance Characterization of Semi-Flexible Pavements"

_coatings, doi:10.3390/coatings12020241_

Round 1
Reviewer 1 Report
The work submitted for review concerns the ever-present problem of the quality and durability of roads. The asphalts used for a long time are subject to continuous modifications in order to improve or better to enhance their quality. The submitted work is a continuation of the previous one. The paper is well written and contains all the essential elements. The authors introduce the reader to the discussed issue well. The only thing missing is a clearly stated research thesis. It should be deduced from the introduction of the work. The methodology is well described and does not raise objections. Some correction can be made and descriptions of the basic equations can be omitted in the work, referring to the known literature. However, it is not a significant mistake. The work is enriched by photographs. The figures, however, require some correction in style. Tables, on the other hand, are well described and there is no problem with analysis of the data contained in them. Conclusions are correctly formulated.
Author Response
Dear reviewer,
Thanks for your comments. The paper was further examined to eliminate potential error or mistakes. The thesis of this article was added in the introduction section.
Regards,
Qi
Reviewer 2 Report
In this paper the volumetric mix design of SFP was implemented then followed by assessing the mechanistic behavior of SFP including strength, stability, viscoelastic behavior, fatigue and rutting resistance by means a series of laboratory tests. The topic is interesting and the results are valuable in some extent. But there are some points that the authors need to be considered before the publication of the paper. The comments are coming in the following paragraphs:
1-Page 5, line 142: What do you mean by water stability? We do not have water stability in asphalt mixture. If you mean resistance to moisture susceptibility it is better to change it. For explaining and assessing moisture resistance the author can refer to two article that is mentioned in no.6.
2- Page 5, Fhg.4: Marshall Stability test doesn’t seem to be a proper test method to assess any mechanical properties of asphalt concrete or SFP. Why did you select this test?
3-Page 7, line 184-186: It has been stated that “The average flexural strength and failure strain of SFP specimens reached 11.18MPa and 2233ߝߤ, respectively, which was significantly greater than traditional asphalt mixtures”. There is no information in Table 2 about flexural strength and failure strain of the traditional asphalt mixtures. How did you make this comparison?
4-Page 8, line 219-220: “The specimen was loaded at a constant strain of 400me”. Why the fatigue test has been performed in one single strain level? Because we do not know what is happening in the other strain levels. Therefore it is difficult to justify about fatigue resistance of one mixture in just one single strain level.
5-The literature survey in the paper needs to improve by adding some more papers including the following articles:
- Investigation of different test methods to quantify rutting resistance and moisture damage of GFM-WMA mixtures, Construction and building materials, 2017.
-Improving moisture and fracture resistance of warm mix asphalt containing RAP and nanoclay additive, Construction and Building Materials, 2021.
Author Response
Dear reviewer,
Thanks for your comments. The article was further revised based on your advices.
1) Moisture susceptibility was used to replace "water stability". We admit that this phrase was more formal and accurate.
2) Some more literatures including the suggested ones were added in the introduction section.
3) The Marshall stability test. It's true that this test was kind of out-dated in asphalt pavement area. The only reason we used it here was to provide a simple and direct glimpse of the curing effect of cement. This method was meant for qualitative use instead of a rigorous quantitative study. A short paragraph was added in section 2.3 to explain this issue.
4) Flexural strength comparison. The original description was not rigorous, so the paragraph was rewritten and another literature was added to help the comparison. The flexural strength range of dense graded HMA was reported to be 4.5 to 7.5MPa so the conclusion was justified.
5) Fatigue test issue. This one was a little bit intricate. As we see, most asphalt mixture studies tend to employ strain-controlled mode to test fatigue lives. This mode will ensure the failure of the specimens. So testing the specimens at multiple strain levels can give us a fatigue equation, which is usually a straight line in log scale. But for SFP this can be tricky, because the cement changed the nature of asphalt mixture, making the specimen less flexible. Using the strain-controlled mode could greatly underestimate the fatigue resistance. This is likely to invalidate the fatigue equation method, because the test results may tell us that cement lead to sooner fatigue failure, which is misleading. So we may need a stress-controlled fatigue test here. But the specimen would not be guaranteed to fail in stress-controlled mode due to the existance of fatigue limit, so comparison with other HMA materials was still difficult. Meanwhile, the significant variance of fatigue test made the situation even worse. To conclude, the best solution here is to perform a comprehensive stress-controlled test along with a probabilistic study, such as survival analysis. But such work is clearly beyond the scope of this paper. Alternatively, we still used strain-controlled mode here. The 400me strain level was chosen because the specimen could fail in an ideal period, not too soon or too late. Fatigue life comparison was made between open-graded asphalt mixture and SFP, which re-validated the effect of cement. But it was also mentioned in the revised manuscipt that comparison between SFP and other asphalt mixture under strain-controlled mode was not justifiable.
Thanks for your comments!
Regards,
Qi
Reviewer 3 Report
As I understand that Semi-Flexible Pavements' high-temperature stability is better than that of asphalt pavement, and it has superior deformation resistance, water damage resistance, and skid resistance. This research paper presents significant breakthroughs in this field. I have no major comments. I suggest a revision of the manuscript for language and grammatical errors.
Minor comments
- Figures captions are too short. Please explain them in detail.
- How your research work is different from previous studies? Highlight your main findings.
- Journals names in the bibliographic section must be abbreviated according to formate.
Author Response
Dear reviewer,
Thanks for your advices. The paper was revised as suggested.
1) The Figure captions were re-edited to better explain the contents.
2) Main findings were highlighted in the summary section.
3) The journals names were updated in the bibliography.
Regards,
Qi
Round 2
Reviewer 2 Report
The all comments have been considered and the corrections have been made thoroughly.
This manuscript is a resubmission of an earlier submission. The following is a list of the peer review reports and author responses from that submission.
Round 1
Reviewer 1 Report
I quite acknowledge the diligent and significance work of the authors. The work presented is relevant and interesting.
Remarks:
1. Give in the abstract and introduction of the paper, the purpose of the research paper.
2. Describe the discussion of Figure 3 in more detail.
3. Add throughout the paper to the term "strength", what strength it is, e.g. line 114, 122, 124, 133, 137, 141, 170, chapter 4 and where not noted.
4. Line 133. Correct the beginning of the sentence.
Author Response
Dear reviewer,
Thanks for your advices. The following revisions were made based on your suggestions:
1) The purpose of this paper was added into the abstract and introduction, as well as the project background.
2) The discussion of Figure 3 was rewritten to explain the findings in detail.
3) We realized the ambiguity caused by the "strength" throught the paper. They were corrected accordingly.
4) The format, spelling and grammer was further examined and revised throughout the article.
Reviewer 2 Report
The article presented for review concerns the current problem of quality and durability of road surfaces. For many years new solutions have been searched for in order to develop a durable asphalt pavement suitable for given conditions. SFP mixes are an example of such a solution. The authors evaluated the quality of an SFP mix composed of two parts, open-graded asphalt mixture and cement grout.
The authors determined the necessary parameters for both mixtures, i.e. void ratio and water-cement ratio, which were 25% and 0.25. They then tested the selected system.
The introduction of the paper is well written, the authors refer to a sufficient number of literature items. In general, the structure of the work allows to understand the tests carried out, but the classical structure - separation of methodology and test results is a better solution. Conclusions correspond to the set goals. The authors include a lot of photos, which makes the work more attractive.
Detailed remarks:
Why did the authors conduct the study using water-cement ratios of 0.25, 0.3 and 0.35? Was it not immediately predictable that a ratio of 0.25 would prove to be the most advantageous?
The introduction of the paper is well written, the authors refer to a sufficient number of literature items. In general, the structure of the work allows to understand the tests carried out, but the classical structure - separation of methodology and test results is a better solution. Conclusions correspond to the set goals. The authors include a lot of photos, which makes the work more attractive.
Detailed remarks:
Why did the authors conduct the study using water-cement ratios of 0.25, 0.3 and 0.35? Was it not immediately predictable that a ratio of 0.25 would prove to be the most advantageous? Can you provide what more detailed characteristics you used to select these ratios?
Please provide more detailed data of the testing machines used.
The authors give some statistical data, but do not specify what information they are providing and there is no information about the number of specimens tested in each study.
The last section "Discussion and conclusions" is a bit different from the accepted templates. In the Discussion chapter we expect references to other authors' research. Here the Authors rather briefly discuss what they did. Drawn conclusions, however, are correct and, in my opinion, these parts should be separated or the chapter should be called Summary.
Author Response
Dear reviewer,
Thanks for your review and advice. We made the following revisions to this paper based on your suggestions.
1) An explanation regarding the the selection of water-cement ratio was added. Basically this is a tradeoff between strength and workability. With higher w-c ratio, we got specimens with lower strength, which is bad; but such low strength material could be easier to flow and thus more convenient for construction. So we have to make comparison to see which is the best w-c ratio.
2) The test setup and procedure was explained in detail in Section 2.
3) The fatigue test results involved certain statistical analysis. This part was updated, including a normality test to validate if the repeated test results followed a normal distribution. Clearly the variance of fatigue test was significant, so describing the fatigue life with a simple number turned out to be fruitless. Thus we believe that treating it as a random variable and representing it with a normal distribution is a better choice.
4) We agreed that the title "Discussion and conclusions" was not a proper title. It is now replaced with "Summary".
Reviewer 3 Report
The paper presents the laboratory characterization of grouted macadam (or semi-flexible pavement material).
The study does not present any element of novelty as compared to the current state of the art:
- there is no comparison with a reference asphalt mixture;
- no variables related to the mechanical behaviour of grouted macadam are studied (e.g. different properties of the open-graded asphalt mixture, different properties of the cement, different aging levels)
- the adopted laboratory testing methods are not advanced and the analysis of the results is too superficial.
In addition, the paper is not well written:
- the structure of the paper is very confusing. A scientific paper is usually prepared considering a “Materials and methods” section and then a “Results and analysis” section, which is not the case of this paper;
- the quality of all photographs is very bad and most of them are not really necessary in a scientific paper, because they show well known equipment/test setups (e.g. Figures 2, 5, 11, 12)
Therefore, the authors are invited to significantly extend the investigation before resubmitting an improved version of the paper, prepared more carefully.
Author Response
Dear reviewer,
Thanks for your review and comments. The following revision was made based on your suggestions.
1) The project background of this paper is added in the introduction section. Basically the work of this paper serve as support to the pavement construction of Bus Rapid Transit in Chengdu, China. So this is more like a case study.
2) The concerns of test method and conditions are further explained in detail in Section 2. It's quite true that the test methods are classic and frequently used in asphalt mixture studies. But when they are used on SFP material, the purpose of the test sometimes can be different. For example, the Marshall stability test we used here was not for the Marshall mix design, since the mix design was already done in volumetric approach. Marshall stability test here served as an efficient way to see how the strength of SFP was growing with curing time. Another consideration is that these classic test methods are simple enough that they can be easily repeated by the technicians in the field, which is quite advantageous in the practice of construction. Some comparison with traditional HMA was also added. Due to the strength provided by cement, SFP can be 10 times stronger than HMA so we don't think such competition is fair.
3) The organization of section 2 and section 3 was updated. The tests performed in section 2 and section 3 are quite different. In section 2, the tests are classic and would produce empirical results regarding strength and stability issues. So the application of these tests stays in the mix design phase. But in section 3, dynamic modulus test and HWT test are performance-oriented test and will directly contribute to the mechanistic-empirical approach in structrual design phase. Thus we believe that mixing these two types of tests is not a good idea. However, we do agree that in each section, the "material-method-results-analysis" order should be followed.
Round 2
Reviewer 3 Report
The authors made only minor changes to the paper. In my opinion, the paper still presents serious deficiencies and cannot be published as it is.
The authors made only minor changes in the introduction and in the description of the test setups, without addressing my previous comments: - the experimental investigation was not extended (e.g. by including the study of some variables and/or additional and more advanced tests); - the analysis of the results was not deepened; - the structure of the paper is unchanged; - the low quality photographs were not changed/removed. Therefore, in my opinion, the paper cannot be published as it is.Author Response
Dear reviewer,
Thanks for your advices. The manuscript was further revised based on your comments in the following aspects:
1) In the experimental investigation part, MMLS3 accelerated wheel tracking tests were added in section 3.2.4, along with performance comparison against other commonly used asphalt mixtures.
2) In the mix design part, the determination of the selected gradation was briefly explained. The gradation of SFP was based on permeable pavements and further adjusted the void ratio to achieve 25%. Due to the special mix design approach of SFP, a multi-variable analysis used in material study is not quite suitable here. As we can see, once the void ratio was known, the structure was determined since all the voids would be filled with cement, and the strength mostly comes from the cement. As a result, the objective of asphalt mix design here is to maximize the void ratio. In practice, the 25% void ratio was the limit we got using a high viscosity asphalt, and using other kinds of asphalt will not contribute to a better mix design. The strength of the cement was primarily determined by the water-cement ratio as well as curing time, which was illustrated in Section 2.2. Admittedly, testing SFP composed of different asphalt, gradation, or cement with different addictive would make the research more rigor and persuasive, but that’s beyond the scope of this paper, and should be presented in a comprehensive research report instead of a concise paper. Thus, we chose to focus on one specific SFP material, and study it using multiple test methods covering strength, fatigue, modulus and rutting.
3) the structure of the manuscript was changed. Section 2 was about mix design, and the tests in section 2.3 was part of the mix design validation. Section 3 presented pavement performance-oriented tests following a “method-result & analysis” order.
4) The blur figures were removed or replaced as requested.